# Risk factors for adverse pregnancy outcomes in Chinese women: a meta-analysis

Yiping Huang, Junbi Xu, Bin Peng and Weiying Zhang

Department of Gynaecology and Obstetrics, The Fourth Affiliated Hospital, Zhejiang University School of Medicine, Yiwu, China

## ABSTRACT

**Objective:** This study examined the associated risk factors of adverse pregnancy outcomes among Chinese females and furnished some fundamental principles and recommendations for enhanced prevention of adverse pregnancy and preservation of women's well-being.

**Methods:** A systematic review was conducted by retrieving the MEDLINE (The National Library of Medicine), Embase, PubMed, and Cochrane databases. The relevant risk factors for adverse pregnancy in Chinese women were retrieved from May 2017 to April 2023. Use Review Manager for data analysis. Calculate the merge effect based on data attributes using mean difference (MD) or odds ratio (or) and 95% confidence interval (CI). The meta-analysis was registered at INPLASY (International Platform of Registered Systematic Review and Meta-analysis Protocols, 202340090).

**Results:** A total of 15 articles were included, with a total of 946,818 Chinese pregnant women. Moreover, all the literature was scored by the NOS (Newcastle-Ottawa Scale), and all literatures were ≥7 points, which were evaluated as high quality. There are seven risk factors related to adverse pregnancy in Chinese women: parity, pregnancy frequency, education level, smoking, gestational diabetes, gestational weeks, and age. Moreover, the main risk factors for adverse pregnancy are pregnancy frequency, education level, gestational diabetes mellitus, and age.

**Conclusion:** The pregnancy frequency, education level, gestational diabetes mellitus, and age were significantly associated with the adverse pregnancy in Chinese women, whereas gestational weeks, smoking, and parity had no significant effect on adverse pregnancy.

## INTRODUCTION

Expected adverse pregnancy consequences comprise preterm labor, macrosomia, neonatal low birth weight, stillbirths, abortions, and placental abnormalities. These outcomes significantly impact the physiological and psychological well-being of both pregnant women and neonates and have garnered worldwide attention (*Chawanpaiboon et al., 2019*; *Muglia et al., 2022*). Infants born in an adverse prenatal environment are also prone to nerve impairments, respiratory ailments, visual and auditory impairments, future

Corresponding author
Yiping Huang,
hanhanlian15@163.com

developmental delay, cognitive impairments, and cerebral palsy (*Yamazaki et al., 2018*; *Olsen et al., 2022*). Hence, there exists apressing requirement to mitigate the possibility of unfavorable delivery consequences *via* regulation of risk determinants for adverse pregnancies or surveillance and intervention in expectant women at high risk. Empirical research has demonstrated that advanced maternal age, high body mass index, length of gestation, multiparity, multifetal gestation, gestational diabetes mellitus, and level of education are all associated with adverse pregnancy outcomes (*Lomelino, Luísa & Anabel, 2019*; *Wado et al., 2019*; *Aydın, Ünal & Özsoy, 2021*; *Griege et al., 2018*; *Lufele et al., 2017*; *Szmuilowicz, Josefson & Metzger, 2019*; *Abraham et al., 2017*).

The number of pregnancies is one of the risk factors for pregnancy (*Çelik & Güneri, 2020*; *Jurjević & Telarović, 2022*). The existing literature has confirmed that primiparous women are risk factors for placental malaria and obstetric diseases. The parity model of malaria susceptibility in malaria-endemic areas suggests that primipara and secondary parturients (to a lesser extent) are more affected than multiple births (*McGregor, 1987*).

With the increasing prevalence of obesity among women of childbearing age, the incidence of gestational diabetes will continue to rise, leading to a vicious cycle of diabetes between mother and child. Studies have shown that offspring of mothers with mild, untreated diabetes have increased rates of obesity and disorders of glucose metabolism around adolescence (*Franzago et al., 2019*).

The decision to delay childbearing involves many social factors. Hope for career success, equal opportunities in the job market and financial stability are main concerns for most women today. In many countries, the average age of a mother's newborn has been rising steadily to 57 years, and the age of childbearing for women worldwide has been pushed back to more than 35 years (*Adashi & Gutman, 2018*; *Li et al., 2022*). The physiological and pathological changes of female reproductive organs and their inherent diseases can lead to adverse pregnancy outcomes. In addition, advanced maternal age is itself an independent risk factor (*Lomelino, Luísa & Anabel, 2019*).

Studies from both developing and developed countries have shown that higher education can prevent early and unwanted pregnancies. Education and family wealth were related to early childbearing in the five countries studied. Adolescents with secondary education and above have significantly lower rates of pregnancy (*Wado et al., 2019*). Studies have found that smoking during pregnancy was associated with reduced fetal size after the first trimester, especially head and femur length. These effects may be mitigated if the mother stops smoking or reduces smoking during pregnancy (*Abraham et al., 2017*).

This research systematically reviewed the risk factors associated with adverse pregnancy outcomes in Chinese women in recent years. The interrelated variables were methodically synthesized, and adverse pregnancy consequences were investigated *via* meta-analysis to direct prenatal and perinatal healthcare and expedite the consequent formulation of suitable interventions to diminish the incidence of adverse pregnancy outcomes.

# MATERIALS AND METHODS

## Search strategy and selection criteria

A systematic review was conducted by retrieving MEDLINE (The National Library of Medicine, Bethesda, MD, USA), Embase, PubMed, and Cochrane databases. The relevant risk factors for adverse pregnancy in Chinese women were retrieved from May 2017 to April 2023. The search keywords were as follows: "adverse pregnancy," "risk factors," "Chinese women," "adverse pregnancy outcomes," and the appropriate combination of the operators "AND," "OR," and "NOT." The languages included in the study were limited to English. The references from relevant researches are also reviewed. We performed research selection through a series of consecutive stages, including duplicate checking using Endnote software, titles and abstracts screening, and full-text article selection according to the eligibility criteria. These processes were conducted independently by two investigators (Junbi X and Bin P). Conflicts were handled by consensus, and an adjudicator (Yiping H) was consulted when necessary. The Meta-analysis was registered at INPLASY (International Platform of Registered Systematic Review and Meta-analysis Protocols, 202340090). The study was approved by the Institutional Review Board and Research Ethics Committee of the Fourth Affiliated Hospital, Zhejiang University School of Medicine.

## Eligibility criteria

Inclusion criteria: (1) The scope of the study included case-control trials that focus on adverse outcomes during pregnancy; (2) the participants under scrutiny were pregnant women from China; (3) the research took into account academic papers published between May 2017 and April 2023; (4) academic articles selected for review are in English only; (5) various risk factors were analyzed, including one or more of the following: the delivery of a pregnant woman, the total number of previous successful deliveries, level of education, gestation period, the prevalence of gestational diabetes mellitus, history of tobacco use, and maternal age. Exclusion criteria: (1) Meta-analysis, dissertation; (2) non-Chinese women are the subjects of study; (3) non-English literature; (4) when there are overlapping periods in dual or multiple studies reported by the same institution, including the most recent one, the others are excluded; (5) no full text, incomplete data, low quality and repetitive articles were unavailable.

## Document selection

The research was independently evaluated by two reviewers, who selected the paper based on its title, abstract, and full text. Disagreements are resolved through the process of achieving consensus. A systematic review of previously published articles and references was conducted to identify additional relevant literature. Standardized forms were utilized to collect required parameters, which relate to basiccharacteristics of the study and population, such as author information, year of publication, sample size, and age.

### Quality evaluation

The assessment of bias risk was conducted for each study that was enrolled in accordance with the Cochrane Handbook for Systematic Reviews of Interventions (*Higgins et al., 2011*). Each study was evaluated based on six criteria: randomization, hidden allocation, blinding of participants and people, blinding of results, incomplete outcome data, and other risk biases. RevMan 5.4 statistical software was used to summarize the bias risk assessment results and draw the bias risk summary map. Furthermore, the quality of evidence was conducted using the Grading of Recommendations, Assessment, Development, and Evaluations, which takes into account the risk of bias, inconsistency, indirect risk, imprecision, and publication bias. GRADE (Grading of Recommendations Assessment, Development and Evaluation) system was used for evaluation. The evidence was classified as a high, medium, low, or very low quality based on these factors. The methodological quality of the observational studies included in the analysis was evaluated using the Newcastle-Ottawa scale (NOS), as per the established protocol (*Wells, Shea & O'Connell, 2011*). The NOS "star system" is a rating scale that ranges from 0 to 9. Research studies that achieve a rating of seven or more stars are classified as high quality. Sensitivity analyses were conducted to evaluate the influence of including studies with a significant risk of bias on the overall results. Two researchers carried out the assessment autonomously, and any conflicting perspectives were reconciled through discussions among the three researchers.

### Statistical methods

Information was obtained from each of the enrolled studies using predetermined tables and then condensed into tables that included (i) article identification, (ii) methodology, (iii) outcome measures, and (iv) outcomes. The statistical significance of the result was commonly determined by a $P$-value < 0.05 or a 95% CI (confidence interval) that did not include the effect measure ratio (relative risk (RR), odds ratio (OR), or hazard ratio (HR)). If there is no ratio or ratio effect measure in the report, but the raw data is obtainable, the author calculates the effect measure. A meta-analysis was performed on datasets from at least three studies that focused on related topics of pregnancy-related adverse outcomes.

The statistical analysis was conducted utilizing RevMan 5.3. The outcomes were presented as the MD (mean deviation) or OR with a 95% CI. A $P$-value < 0.05 was deemed statistically significant unless otherwise specified. Furthermore, the Q test and $I^2$ statistical approach were employed to measure heterogeneity. The fixed-effect model was employed when the heterogeneity test indicated no significant differences ($P > 0.05$ and $I^2 < 50\%$). If the data exhibited heterogeneity, a random effects model is employed.

## RESULTS

### Results of document retrieval

Following a systematic database search, a total of 620 studies were initially identified. Subsequently, the titles and abstracts of these studies were meticulously screened, and 315 duplicate articles were eliminated, resulting in a final set of 305 articles, including reviews, case reports, letters, and unrelated studies, which were excluded from further analysis.

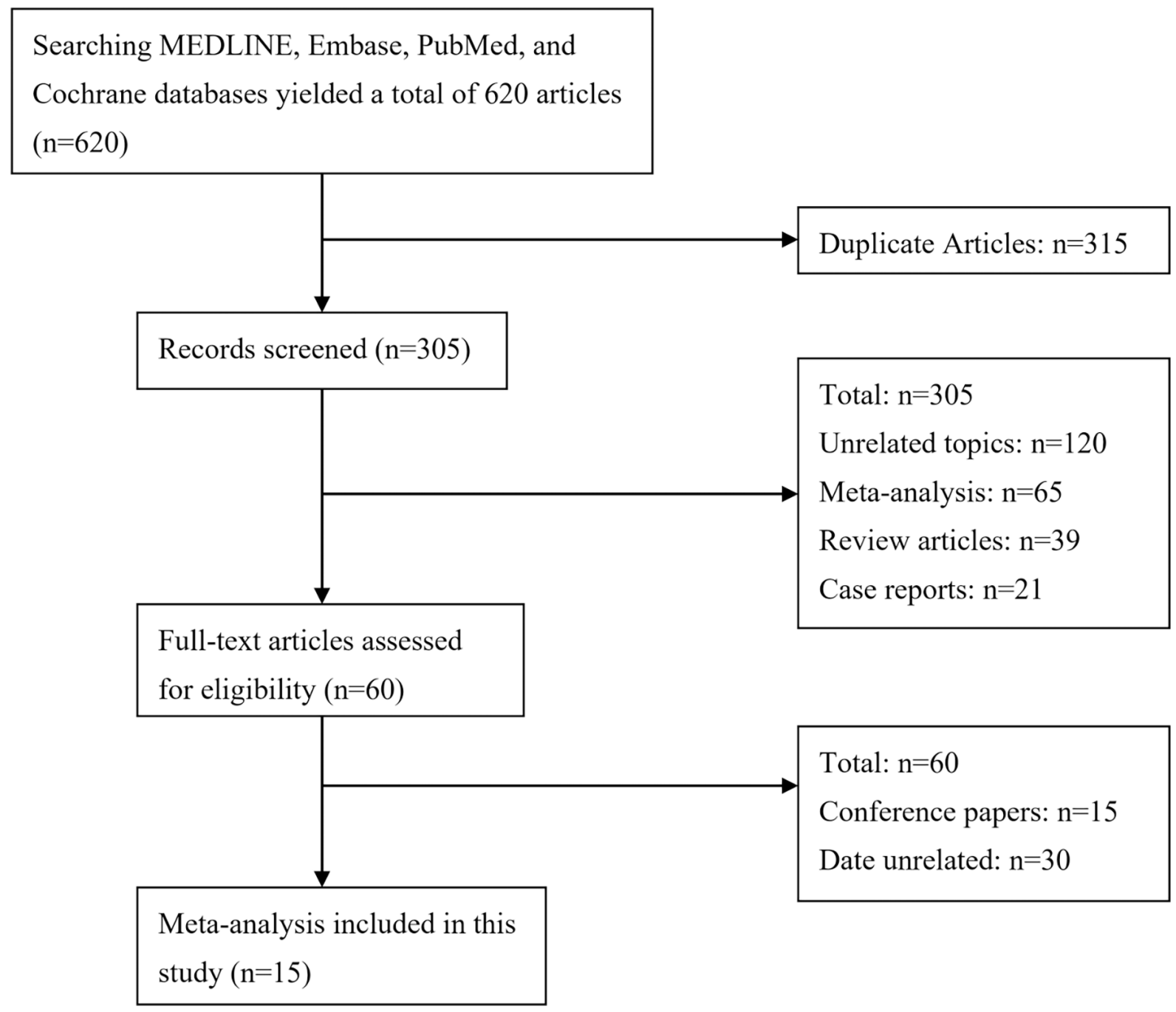

**Figure 1 Flow chart of the literature search strategy.**

The complete text of the remaining 60 articles underwent meticulous evaluation, and a total of 15 studies that met the inclusion and exclusion criteria were ultimately incorporated into the meta-analysis (*Liu et al., 2021*; *Jing et al., 2018*; *Jia et al., 2020*; *Bi, Zhang & Wang, 2020*; *He et al., 2018*; *Zhao et al., 2021*; *Liu et al., 2020*; *Zhang & Zhao, 2021*; *Su, Chen & Huang, 2019*; *Su et al., 2020*; *Lin et al., 2021*; *Chen et al., 2021*; *Chen et al., 2020*; *Lin et al., 2020*; *Sun et al., 2020*). This analysis comprised 946,818 pregnant women, and Table 1 displays the fundamental characteristics of the studies that were incorporated. Figure 1 displays the flow chart of the document selection procedure.

## Basic characteristics

The research articles were published from 2017 to 2022 in English. In the 15 studies, all of the subjects were Chinese women. The basic characteristics of the 15 included studies are shown in Table 1.

## Methodological evaluation and outcome of bias assessment risk

Of the 15 included studies, one did not report a randomized sequence generation method, one did not provide allocation concealment information, and ten were double-blind studies. Data from two studies were not reported in detail. The results of the quality evaluation of the included studies and the bias risk assessment are shown in Figs. 2 and 3.

**Table 1** Baseline characteristics of included studies.

| First author | Year | Number | | Age | | BMI | | Type of study | Risk factors | NOS |
|---|---|---|---|---|---|---|---|---|---|---|
| | | Expermental | Control | Expermental | Control | Expermental | Control | | | |
| Chen-Ning Liu | 2021 | 506 | 32828 | 187 (>35) | 7419 (>35) | 111 (>25) | 6482 (>25) | Cohort study | 2,5,7 | 7 |
| Lin Jing | 2018 | 157 | 519 | 32.8 ± 4.2 | 32.2 ± 3.9 | | | Retrospective study | 7 | 8 |
| Lu Jia | 2020 | 2,988 | 113054 | 8 (>35) | 271 (>35) | | | Cohort study | 1,2,3,4,7 | 7 |
| Shilei Bi | 2021 | 2,651 | 466 | 8 (>35) | 271 (>35) | | | Cohort study | 1,3,4,5,7 | 7 |
| Yang He | 2018 | 270 | 176 | 24.5 ± 4.0 | 24.7 ± 5.6 | | | Cross-sectional study | 3,6 | 7 |
| Nan Zhao | 2021 | 126 | 310 | 49 (>35) | 128 (>35) | 32 (>25) | 74 (>25) | Case-control study | 2,3,5,6,7 | 7 |
| Lu Liu | 2020 | 444 | 1033 | 30 | 31 | | | Cohort study | 2,4,5,6 | 7 |
| Hehua Zhang | 2020 | 502 | 1846 | 31.6 | 30.3 | 23.9 | 22.2 | Cohort study | 1,3,7 | 7 |
| Wei-juanSu | 2019 | 6,982 | 8874 | | | 26.97 ± 1.97 | 23.82 ± 0.56 | Population-based study | 2,3,5 | 7 |
| XiujuanSu | 2020 | 440 | 8336 | 30.4 ± 3.7 | 30.2 ± 3.6 | | | Retrospective study | 2 | 7 |
| Li Lin | 2021 | 318,424 | 427986 | 30.86 ± 4.43 | 26.46 ± 3.77 | | | Retrospective study | 3 | 7 |
| Xi Chen | 2021 | 464 | 6038 | 27.9 ± 3.6 | 28.3 ± 3.5 | 20.8 ± 2.8 | 20.8 ± 2.8 | Cohort study | 2,3,5 | 8 |
| Jiebing Chen | 2019 | 83 | 85 | 28 | 28 | | | Observational study | 2,3,5,6 | 7 |
| Xueyan Lin | 2018 | 139 | 142 | 31.4 ± 4.9 | 31.8 ± 5.1 | 25.4 ± 3.4 | 25.9 ± 3.7 | Randomized study | 1 | 8 |
| Yin Sun | 2020 | 59 | 2292 | 8 (>35) | 271 (>35) | | | Cohort study | 3,7 | 7 |

**Note:**
1, Gravidity; 2, parity; 3, education level; 4, gestational age; 5, gestational diabetes mellitus (GDM); 6, smoking; 7, age.

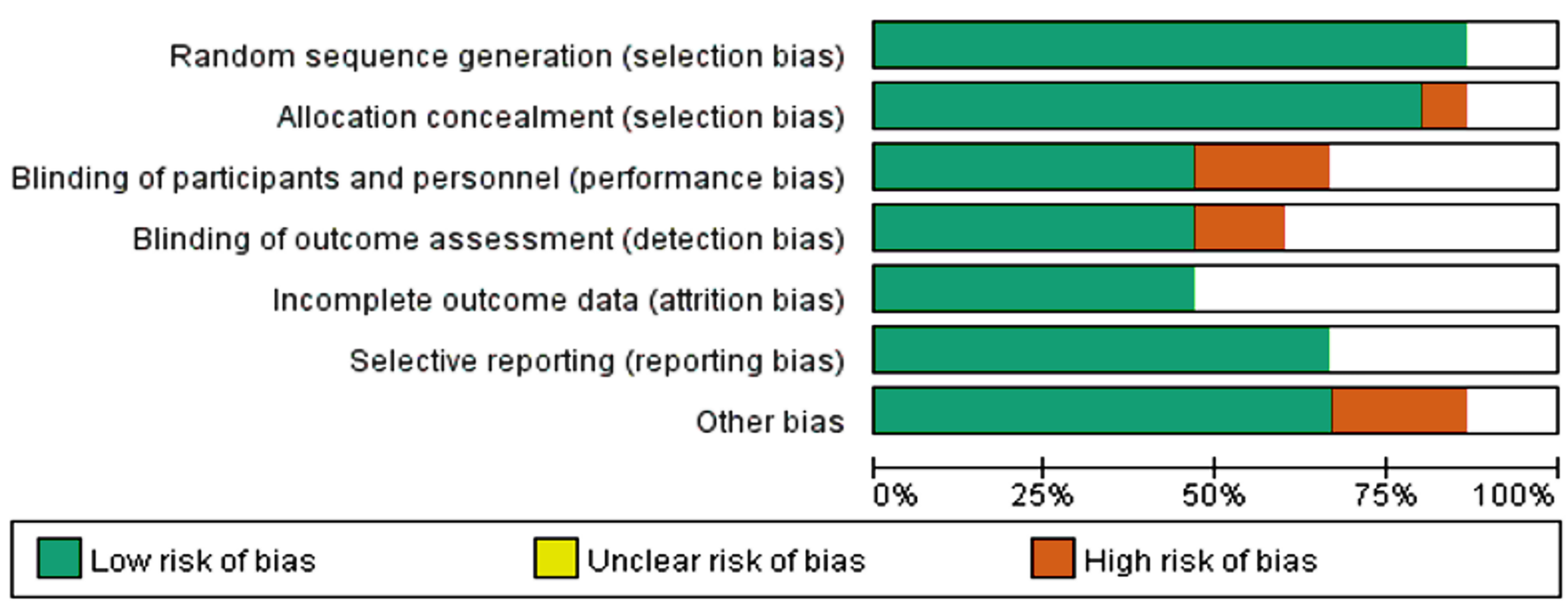

**Figure 2** Quality assessment of included studies.

## Outcome of meta-analysis

### *Meta-analysis of the influence of pregnancy frequency on adverse pregnancy in Chinese women*

A total of four articles were included in the related analysis of the effect of the number of pregnancies on adverse pregnancy. A total of 5,746 subjects were included in the study. The experimental group consisted of 3,292 subjects with fewer than two pregnancies, and the control group consisted of 2,454 subjects with fewer than two pregnancies.
The heterogeneity of the four articles was significant ($P$ = 0.14, $I^2$ = 49%). Fixed-effect model and OR value were used as the combined effect value. As indicated in Fig. 4,

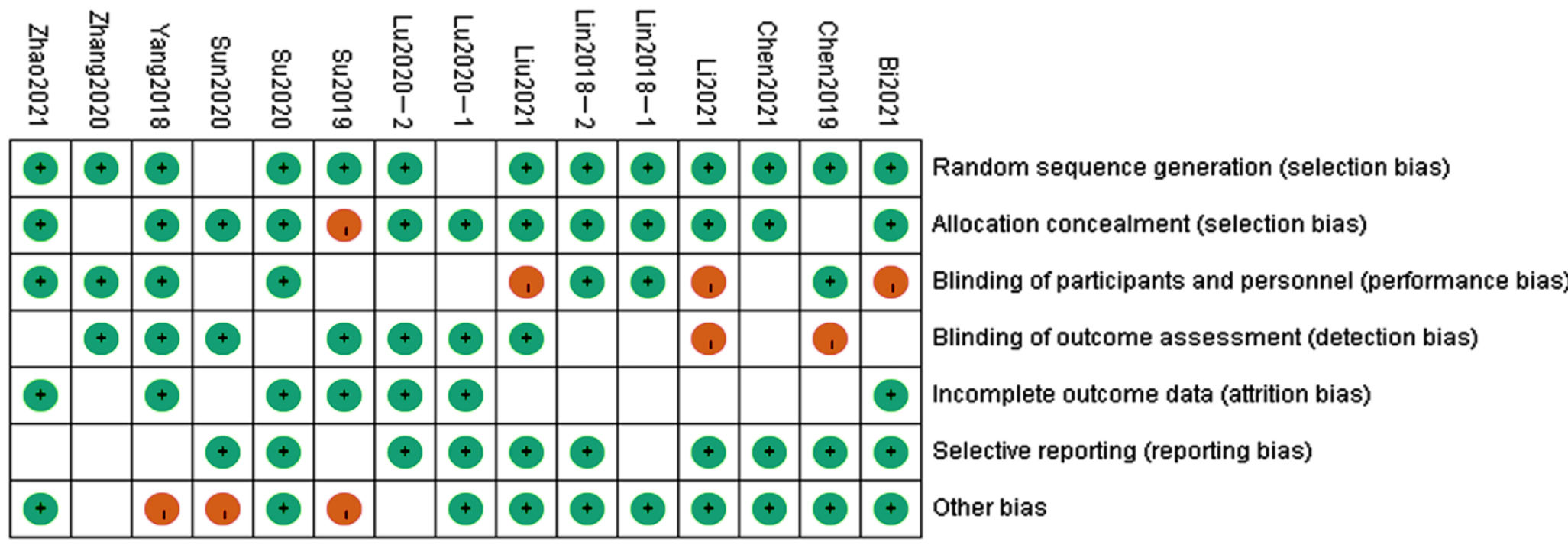

**Figure 3** **Quality assessment of included studies.**

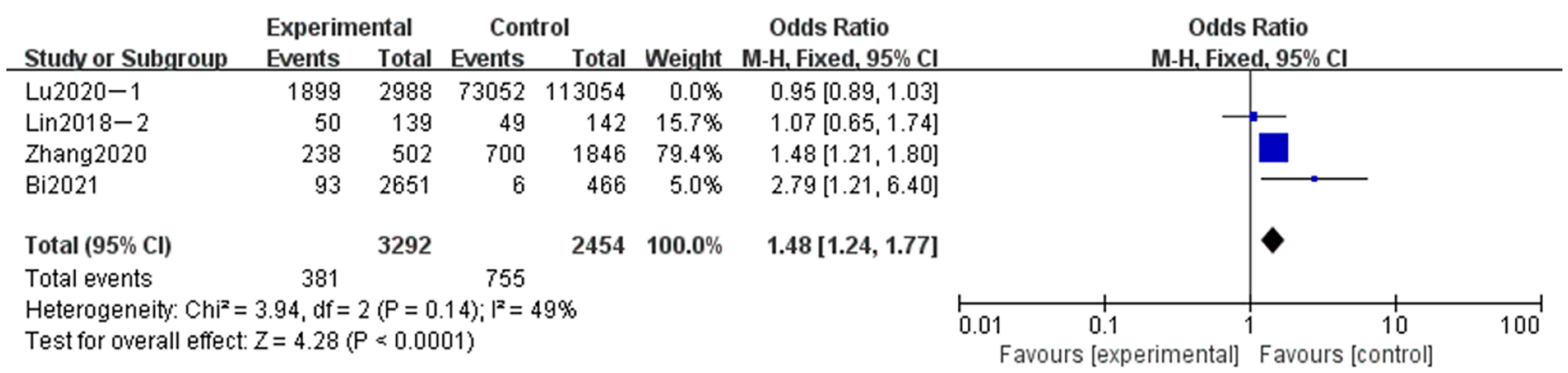

**Figure 4** **Meta-analysis of the influence of gravidity on adverse pregnancy in Chinese women.**

pregnancy frequency was a related risk factor for adverse pregnancy in China (OR = 1.48, 95% CI [1.24–1.77], Z = 4.28, $P < 0.0001$).

### *Meta-analysis of the influence of parity on adverse pregnancy in Chinese women*

A total of eight articles were included in the related analysis of the effects of the number of delivered babies on adverse pregnancy outcomes, covering 182,591 subjects, 170,558 in the experimental group, and 12,033 in the control group. The results showed that the heterogeneity had little change ($P < 0.00001$, $I^2 = 95\%$). Random effect model and OR value were used as the combined effect value. As indicated in Fig. 5, the number of delivered children did not affect adverse pregnancy (OR = 1.46, 95% CI [0.98–2.18], Z = 1.86, $P = 0.06$).

### *Meta-analysis of the influence of education level on adverse pregnancy in Chinese women*

A total of 10 articles were included in the analysis of the effect of educational level on adverse pregnancy, including 893,676 subjects, 332,549 subjects with a bachelor's degrees or above in the experimental group, and 561,127 subjects with a bachelor's degree or above in the control group. The results showed that the heterogeneity had little change ($P < 0.00001$, $I^2 = 98\%$). Random effect model and OR value were used as the combined effect value. As indicated in Fig. 6, education level was the related risk factor of bad pregnancy (OR = 0.57, 95% CI [0.44–0.72], Z = 4.6, $P < 0.00001$).

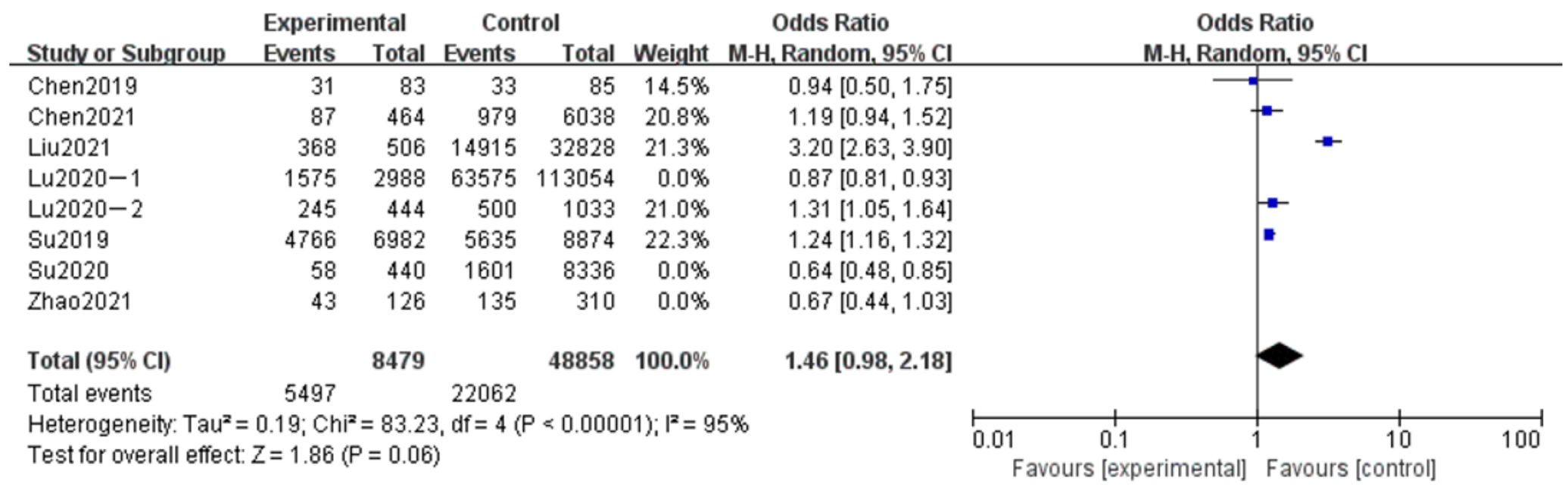

**Figure 5** **Meta-analysis of the influence of parity on adverse pregnancy in Chinese women.**

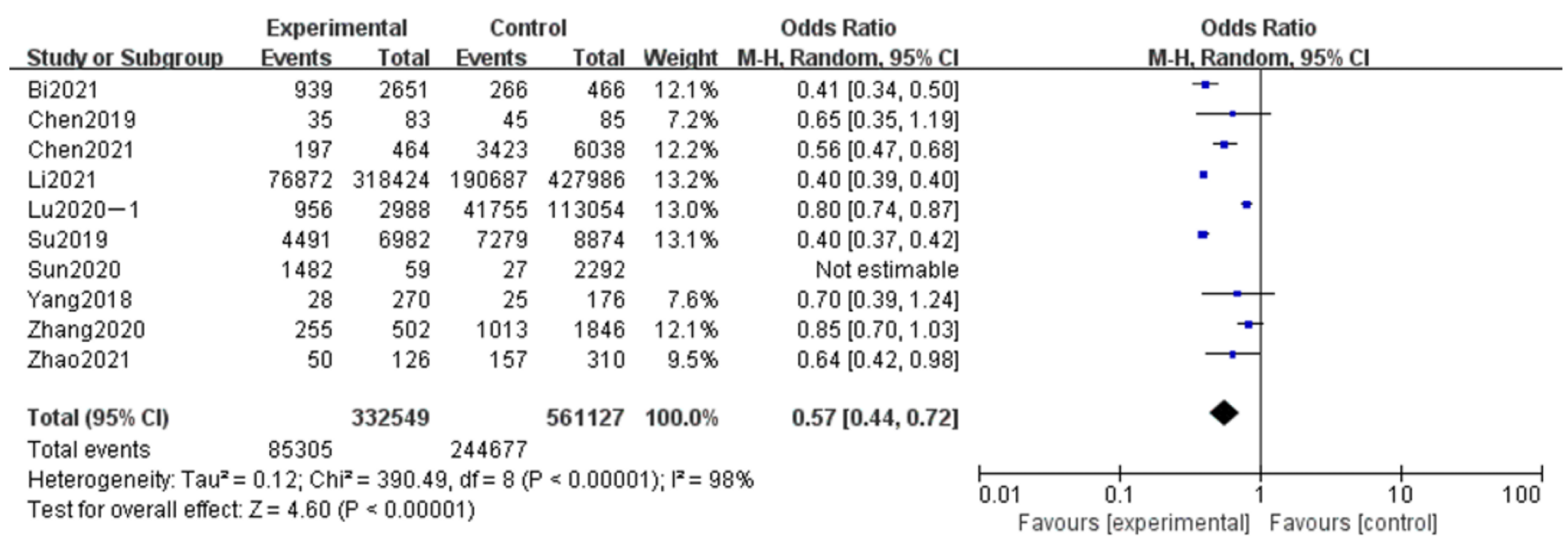

**Figure 6** **Meta-analysis of the influence of education level on adverse pregnancy in Chinese women.**

### *Meta-analysis of the influence of gestational weeks on adverse pregnancy in Chinese women*

A total of three articles were included in the related analysis of the effect of gestational weeks on adverse pregnancy, covering 4,594 subjects, 3,095 in the experimental group, and 1,499 in the control group. The results showed that the heterogeneity had little change ($P < 0.00001$, $I^2 = 99\%$). Random effect model and OR value were used as the combined effect value. As indicated in Fig. 7, gestational weeks did not affect adverse pregnancy (OR = 0.22, 95% CI [0.59–0.15], Z = 1.16, $P = 0.26$).

### *Meta-analysis of the influence of gestational diabetes mellitus on adverse pregnancy in Chinese women*

A total of seven articles were included in the related analysis of the effect of gestational diabetes mellitus (GDM) on adverse pregnancy, covering 60,722 subjects, 11,173 in the experimental group, and 49,549 in the control group. The results showed that the heterogeneity had little change ($P = 0.02$, $I^2 = 62\%$). Randomized effect model and OR value were used as the combined effect value. As indicated in Fig. 8, GDM was a related risk factor for adverse pregnancy (OR = 1.22, 95% CI [1.04–1.44], Z = 2.42, $P = 0.02$).

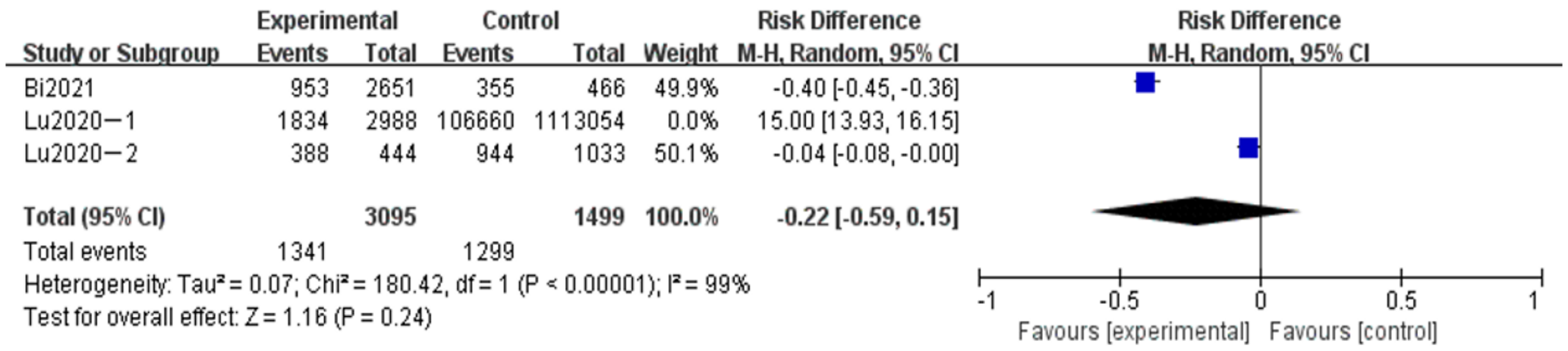

**Figure 7** **Meta-analysis of the influence of gestational age on adverse pregnancy in Chinese women.**

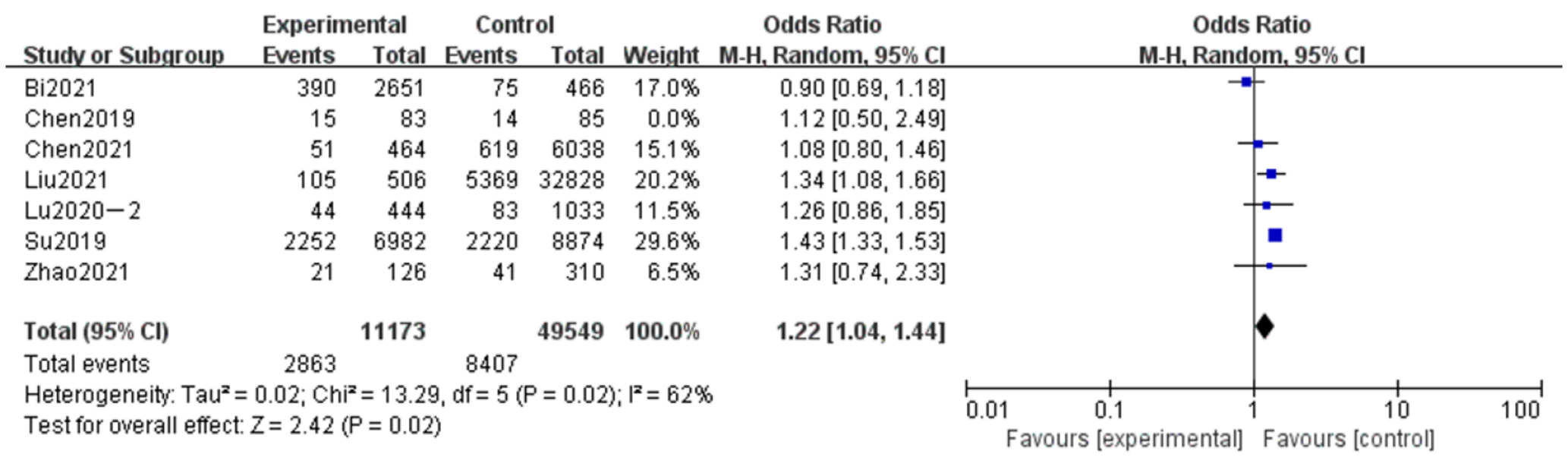

**Figure 8** **Meta-analysis of the influence of GDP on adverse pregnancy in Chinese women.**

### *Meta-analysis of the influence of smoking on adverse pregnancy in Chinese women*

Smoking is defined as smoking during pregnancy by a pregnant woman or a family member living with her (a pregnant woman's exposure to smoking). A total of five articles were included in the related analysis of the effects of smoking on adverse pregnancy, covering 4,429 subjects, 1,155 smokers in the experimental group and 3,274 smokers in the control group. The results showed that the heterogeneity varied significantly ($P < 0.78$, $I^2 = 0\%$). Fixed-effect model and OR value were used as the combined effect value. As indicated in Fig. 9, smoking did not influence adverse pregnancy (OR = 1.17, 95% CI [0.98–1.40], Z = 1.72, $P = 0.08$).

### *Meta-analysis of the influence of age on adverse pregnancy in Chinese women*

A total of 155,280 subjects were included in five articles concerning the analysis of the effect of age on adverse pregnancy. Six thousand three hundred thirty subjects were over 35 in the experimental group, and 148,950 subjects were over 35 in the control group. The results showed that the heterogeneity had little change ($P = 0.03$, $I^2 = 75\%$). Randomized effect model and OR value were used as the combined effect value. As indicated in Fig. 10, age is a related risk factor for adverse pregnancy (OR = 1.51, 95% CI [1.22–1.87], Z = 3.81, $P = 0.0001$).

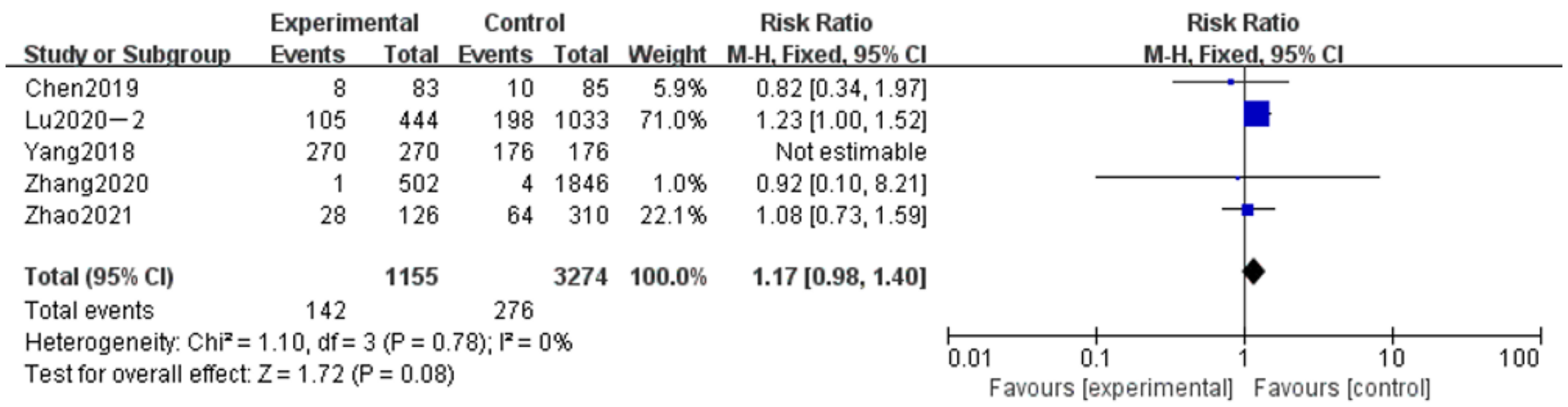

**Figure 9** **Meta-analysis of the influence of smoking on adverse pregnancy in Chinese women.**

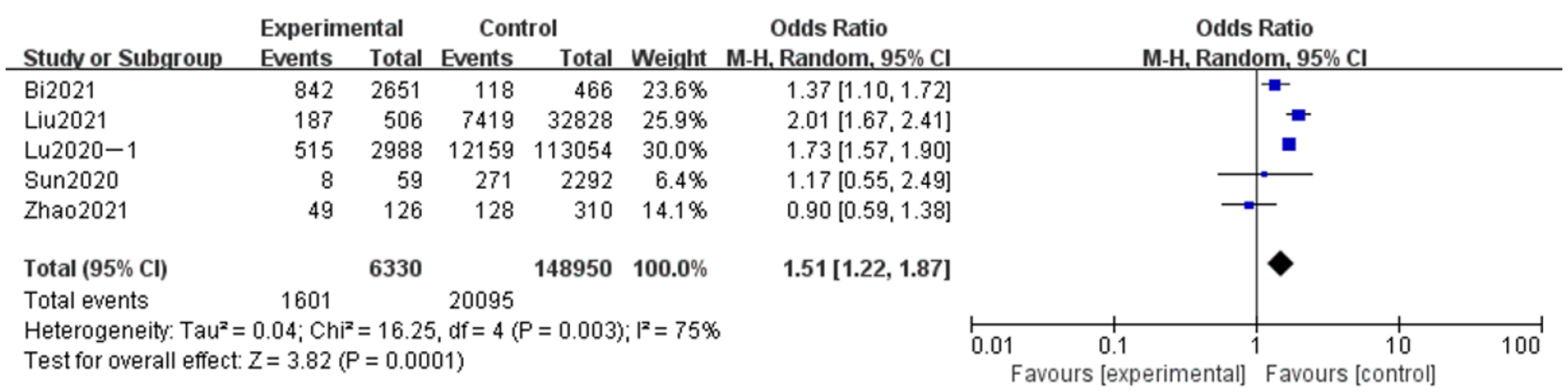

**Figure 10** **Meta-analysis of the influence of age on adverse pregnancy in Chinese women.**

# DISCUSSION

China's population policy has shifted from controlling population size to improving population quality. With the liberalization of the three-child policy, how to improve prenatal and postnatal care to reduce the occurrence of unhealthy pregnancies is a new problem that needs to be solved. Early detection of adverse pregnancy risk factors, pay attention to etiology management, pregnancy guidance and perinatal care, to avoid the occurrence of adverse pregnancy.

The results showed that maternal age, pregnancy frequency, education level and gestational diabetes mellitus were the related risk factors for adverse pregnancy in Chinese women.

It is generally believed that the aging of the placenta in elderly pregnant women is accelerated, and the transport of nutrients and vascular function are changed, resulting in uterine and placental perfusion deficiency, diabetes, hypertension, and other chronic diseases. Placental dysfunction is a potential factor for adverse pregnancy in elderly pregnant women. In other published studies, the stillbirth rate in elderly parturients is higher regardless of the fetus number (*Nyrhi et al., 2023*). Older pregnant women are often considered high-risk (even those without known risk factors). This generalization reduces the clinical threshold of obstetric intervention (*Lomelino, Luísa & Anabel, 2019*). A meta-analysis study have suggested that advanced maternal age is a risk factor for caesarean section, and analyzed its possible biological basis: with advanced age, poor progression and prolonged labor, impaired uterine muscle contractions and dystocia are the most commonly discussed causes (*Pinheiro et al., 2019*). Research has investigated the various factors linked to teenage pregnancy, encompassing multidimensional factors at the

personal, relationship, family, and structural levels. Based on the systematic evaluation, it has been consistently observed that teenage pregnancy is correlated with poverty and lower levels of education, among other factors at varying levels. Early marriage, early childbearing, and successful transition from adolescence to adulthood persist because of poverty, lack of education and gender inequality (*Serván-Mori et al., 2022*; *Yakubu & Salisu, 2018*). *Shri et al. (2023)* findings from multivariable logistics regression analysis show that respondent's level of education and age at marriage were found to be statistically significantly associated with adverse pregnancy outcome. The most prevalent complication during pregnancy is gestational diabetes mellitus. It is correlated with adverse outcomes for mother and newborns. Regulating blood glucose levels in expectant mothers diagnosed with gestational diabetes mellitus has the potential to decrease the occurrence of adverse outcomes for mothers and newborns (*Zehravi, Maqbool & Ara, 2021*). *Ye et al. (2022)* conducted a meta-analysis in their study to investigate the correlation between gestational diabetes mellitus and adverse pregnancy outcomes. The results of their analysis indicate a significant association between gestational diabetes mellitus and multiple complications during pregnancy. The potential correlation between adverse pregnancy outcomes and various risk factors in Chinese women has yet to be explored. Hence, a pressing clinical requirement exists to investigate the associated risk determinants of adverse pregnancy outcomes in Chinese females and establish fundamental principles and directives for enhanced prevention of adverse pregnancy and preservation of women's well-being. Previous study has indicated that primiparous women had a higher risk of early gestational age and giant babies than those with pregnancy history (*Zhou et al., 2019*). Pregnancy and childbirth will lead to significant physiological changes, including sex hormone levels, oxidative stress levels, and perinatal hemodynamics. Studies have reported that the greater the number of fetuses, the higher the risk of developing metabolic syndrome and diabetes (*Moosazadeh et al., 2020*).

## CONCLUSIONS

This study has the following limitations, and as with other systematic reviews, our review is limited by publication bias. Published studies were summarized, and no grey literature or other sources were searched. It is also possible that articles are not collected for the restrictions on searching policies. Since this review is limited to Chinese studies, articles from other countries are inevitably excluded, and selection bias may exist. Besides, due to the limited number of included literatures, certain factors related to adverse pregnancy may be omitted because few reports on such related factors are not authoritative for meta-analysis. Particular retrospective literature was included, and the quality of the study was uneven. The NOS scores of the 15 studies included in this study were all greater than seven, indicating high study quality, so the results of this study were real and reliable. In summary, pregnancy frequency, education level, gestational diabetes mellitus and age are significantly correlated with adverse pregnancy in Chinese women, while gestational weeks, smoking and parity have no significant influence on adverse pregnancy.

## Funding

The authors received no funding for this work.

## Competing Interests

The authors declare that they have no competing interests.

## Author Contributions

- Yiping Huang conceived and designed the experiments, analyzed the data, authored or reviewed drafts of the article, and approved the final draft.
- Junbi Xu performed the experiments, authored or reviewed drafts of the article, and approved the final draft.
- Bin Peng performed the experiments, prepared figures and/or tables, and approved the final draft.
- Weiying Zhang conceived and designed the experiments, analyzed the data, prepared figures and/or tables, and approved the final draft.

## Human Ethics

The following information was supplied relating to ethical approvals (*i.e*., approving body and any reference numbers):

The study was approved by the Institutional Review Board and Research Ethics Committee of the Fourth Affiliated Hospital, Zhejiang University School of Medicine.

## Data Availability

The raw data is available in the Supplemental File.

## Supplemental Information

Supplemental information for this article can be found online at http://dx.doi.org/10.7717/peerj.15965#supplemental-information.

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
