# Peer review of "Risk factors for adverse pregnancy outcomes in Chinese women: a meta-analysis"

_PeerJ, doi:10.7717/peerj.15965_

## Round 0.1 · original submission · Minor Revisions

Please carefully read the comments and suggestions from the reviewers and provide your point-to-point responses.

Reviewer 1 ·

Basic reporting

Yes

Experimental design

Yes

Validity of the findings

Yes

Annotated reviews are not available for download in order to protect the identity of reviewers who chose to remain anonymous.

Reviewer 2 ·

Basic reporting

no comment

Experimental design

no comment

Validity of the findings

no comment

Additional comments

I'm very glad to review your manuscript. We found that the author explained the updated meta-analysis of risk factors associated with adverse pregnancy in Chinese women. However, during the trial, we found some problems that need to be modified by the author.

1.The abbreviations appearing for the first time in the abstract and manuscript need to be defined. Please check the full text.
2.Cite others’ literature and compare it with this study to explore its possible mechanisms in the section of Discussion.
3. Please elaborate on the core differences between the meta-analysis you have done and the published meta-analysis with the similar name.
4.In the past five years, the citations of relevant literature should exceed 80%.
5. Are there clear conceptual definitions for the key variables, and are the variables available from the primary studies appropriate given the conceptual definitions above?
6. Are the methods used to extract data from studies justifiable, clearly documented, and repeatable?
7. Please pay attention to English grammar, sentence structure and reformat the paper according to the requirement of the journal.

---

## Round 0.2 · accepted · Accept

The authors have addressed the comments from the reviewers and the manuscript may be acceptable for publication at the current stage.